# Therapeutic outcomes with surgical and medical management for primary aldosteronism: protocol for a systematic review and meta-analysis

Aldo Rocca,[1] Eleftheria Gkaniatsa,[2,3] Maria Chiara Brunese,[1] Eva Hessman,[4] Andreas Muth,[5,6] Bright I Nwaru,[7,8] Oskar Ragnarsson,[2,3] Emanuele Bobbio,[9,10] Daniela Esposito [ORCID] [2,3]

EB and DE contributed equally.

For numbered affiliations see end of article.

**Correspondence to**
Dr Daniela Esposito;
daniela.esposito@gu.se

## ABSTRACT

**Introduction** Treatment strategies for primary aldosteronism (PA) include unilateral adrenalectomy and medical treatment with mineralocorticoid receptor (MR) antagonists. Whether these two different treatment strategies are comparable in mitigating the detrimental effect of PA on outcomes is still debated.

**Objectives** The primary aim of this systematic review is to identify, appraise and synthesise existing literature comparing clinical outcomes after treatment in patients with PA.

**Methods and analysis** A systematic and comprehensive search will be performed using PubMed, Web of Science and EMBASE, for studies published until December 2022. Observational and interventional studies will be eligible for inclusion. The quality of observational studies will be assessed using the Newcastle–Ottawa Scale, while interventional studies will be assessed using the Cochrane Effective Practice Organization of Care tool. The collected evidence will be narratively synthesised. We will perform meta-analysis to pool estimates from studies considered to be homogeneous. Reporting of the systematic review and meta-analysis will be in accordance with the Meta-analysis of Observational Studies in Epidemiology Preferred Reporting Items for Systematic reviews and Meta-Analysis guidelines.

**Ethics and dissemination** As this study is based solely on the published literature, no ethics approval is required. This review will aim to provide some estimates on outcomes, including survival, rates of clinical and biochemical control, cardiovascular and cerebrovascular events, as well as data on quality of life and renal function, in patients with PA treated surgically or with MR antagonists. The study findings will be presented at scientific meetings and will be published in an international peer-reviewed scientific journal.

**PROSPERO registration number** CRD42022362506.

## STRENGTHS AND LIMITATIONS OF THIS STUDY

⇒ Findings from this systematic review will provide a comprehensive and most contemporaneous synthesis of the underlying evidence regarding long-term outcomes in patients with primary aldosteronism who have been treated with either surgery or mineralocorticoid receptor antagonists.

⇒ This systematic review protocol follows the Preferred Reporting Items for Systematic Review and Meta-Analysis Protocols guidelines.

⇒ The identification of studies from leading medical and public health databases, with no geographical or language limitations, will advance the import of this evidence synthesis across settings.

⇒ The reproducibility of our work is enhanced through a priori outline of the review processes before the actual review starts.

## BACKGROUND

Primary aldosteronism (PA), also known as Conn's syndrome, is the most common cause of secondary hypertension, affecting 3–13% of patients with hypertension and 20% of patients with therapy-resistant hypertension.[1]

PA is characterised by autonomous hypersecretion of aldosterone, causing inappropriate potassium excretion, concurrent sodium and fluid retention, as well as low plasma renin concentration.[1 2] PA is most commonly caused by adrenal adenoma or unilateral or bilateral adrenal hyperplasia.[2] Rare causes of PA include adrenocortical carcinoma and inherited forms of familial hyperaldosteronism.[1 2] Patients with PA are at higher risk of cardiovascular diseases such as myocardial infarction, stroke and atrial fibrillation than patients with essential hypertension also when adjusted for blood pressure (BP).[3 4]

Treatment strategies for PA include unilateral adrenalectomy and medical treatment with mineralocorticoid receptor (MR) antagonists.[1] Minimally invasive adrenalectomy is a safe and feasible procedure and recommended treatment for patients with unilateral PA.[5–7] Although adrenalectomy usually results in biochemical cure, hypertension can persist in up to 60% of patients.[7] Conversely,

MR antagonists (spironolactone or eplerenone) are an option in patients with bilateral PA or when surgery is not feasible.[8] Furthermore, medical treatment is preferred in case of uncertain lateralisation, especially at centres where adrenal venous sampling is not available or has been unsuccessful.[9]

Evidence is still limited on whether different treatment strategies mitigate the detrimental effect of PA on short-term and long-term outcomes.[10] Some investigators reported similar outcomes between surgical and medical treatment.[11–13] However, other studies showed better control of arterial BP and hypokalaemia, better cardiovascular outcomes, as well as improved quality of life after adrenalectomy than with medical treatment.[5] In a recent meta-analysis, lower risks of all-cause mortality and/or major adverse cardiovascular events were also reported in surgically treated patients compared with those treated with MR antagonists.[14] However, evidence about the effect of adrenalectomy versus MR antagonists on clinical and/or biochemical control as well as quality of life and renal function is still debated. Furthermore, evaluation of effects and side effects of MR antagonist treatment is hampered by patients frequently being undertreated. Thus, it is paramount to identify and synthesise the available evidence to date in order to gain clearer insights into this field.

### Scientific questions and objectives

The aim of this systematic review is to identify, critically appraise and synthesise available data evaluating whether patients with PA who underwent adrenalectomy have higher survival rate than patients treated with MR antagonists.

The secondary aim of this review is to compare rates of cardiovascular and cerebrovascular events (heart failure, myocardial infarction, stroke), successful clinical control (normalised or improved BP, tapering of antihypertensive medications), biochemical control (normalised plasma renin and potassium levels), as well as improved quality of life and renal function between patients who underwent adrenalectomy and those treated with MR antagonists.

### PROJECT DESCRIPTION
### Methods

We will perform a systematic review and meta-analysis according to Preferred Reporting Items for Systematic Reviews and Meta-Analysis (PRISMA) guidelines.[15]

The main study objectives have been formulated according to Patients, Intervention, Comparison, Outcomes model (table 1). The protocol for this review has been reported according to Preferred Reporting Items for Systematic Review and Meta-Analysis Protocols guidelines and has been registered in PROSPERO (registration number: CRD42022362506).

This systematic review was planned in September 2022 and is expected to be finalised in January 2024.

| Table 1 | PICO |
|---|---|
| P | Patients with PA |
| I | Adrenalectomy |
| C | Medical treatment |
| O | All-cause mortality<br>Cardiovascular and cerebrovascular events: heart failure, myocardial infarction, stroke<br>Clinical control: normalised or improved blood pressure, less antihypertensive medication<br>Biochemical control: normalised renin and potassium<br>Quality of life<br>Renal function |

PA, primary aldosteronism; PICO, Patients, Intervention, Comparison, Outcomes.

### Eligibility criteria and information sources

The population of interest will be patients with PA treated with adrenalectomy in comparison with those receiving MR antagonists. Studies that report survival rates and rates of clinical control (ie, normalised or improved BP, tapering of antihypertensive medication), biochemical control (ie, normalised renin and potassium), cardiovascular and cerebrovascular events (heart failure, myocardial infarction, stroke), as well as data on quality of life or renal function will be included.

Data across different age groups, gender and ethnicity will be analysed, and there will be no exclusion by country or language. We will translate studies reported in other languages than English.

Observational studies (cross-sectional, cohort and case–control) and interventional (randomised controlled trials, field trials, community trials, etc) studies will be included.

Exclusion criteria will be: date of publication before 2000, review articles, case reports, editorials, letters and other articles where the calculation of rates of outcomes is not feasible. If an eligible article does not report the rates of the prespecified outcomes, an attempt to retrieve these data will be performed by contacting the study authors.

Search will be conducted in three databases (PubMed, Web of Science and EMBASE) using commonly used terms stated in related literature as well as Medical Subject Headings terms. Search strategies will be focused on three concepts: the population/diagnosis, surgical treatment and medical therapy. The following terms will be used: hyperaldosteronism, hyperaldosteronaemia, aldosteronism, aldosteronaemia, adrenalectomy(-ies), surgery, surgical resection, MR antagonists, spironolactone, eplerenone. All databases will be searched from January 2000 until December 2022. The full search strategy for all databases is shown in online supplemental material.

An additional manual search based on the citations and references for included reports will also be conducted. All reports obtained from the databases will be exported to the reference management software EndNote. The

search result will then be deduplicated, following the Bramer *et al* method.[16]

## Ethical issues

Ethical approval or informed consent is not required since this systematic review will be based only on previously published studies and does not imply any direct contact with individual subjects.

## Selection process

Article screening will be independently performed by authors AR, EG, MCB and DE by evaluating titles and abstracts for eligibility assessment. Relevant reports will be identified and analysed by at least two authors in a blinded process using the web application Rayyan.[17] In case of disagreement, a third author will resolve it. Bibliographies and citations for the eligible articles will be examined to identify any additional relevant report, which was not retrieved in the literature search.

The selected reports will be categorised into three groups: irrelevant, relevant and unsure. The reports categorised as irrelevant by both reviewers will be excluded. The reports categorised as relevant or unsure by at least one reviewer will be retrieved and the full papers evaluated. Then, for all the potentially eligible reports, the full texts will be carefully reviewed and a selection of reports to include will be made by each reviewer. The reports selected by the reviewers will be compared and any discrepancies will be discussed. When an agreement is not reached, another author (OR) will make the final decision.

The search and selection process will be presented in a PRISMA flow chart.

## Data extraction

Relevant information from included studies will be extracted using a data extraction form that will help to standardise the data extraction process. The following data will be collected: study name, name of the first author, year of publication, country where the study was conducted, total number of patients, source from which study subjects were selected, study design, aetiology of PA, definition of diagnostic criteria for PA, outcomes definition, gender, age, ethnicity, incidence with 95% CIs of the outcomes. When possible, missing information will be calculated from the available data. Authors will be contacted by the reviewers if data are inadequate, unclear or missing from the report. At least two reviewers will independently extract data from all studies. Any discrepancies between reviewers in the data extraction will be resolved by either consensus or a third reviewer will arbitrate.

## Quality assessment

Quality of observational studies will be assessed using the Newcastle–Ottawa Scale, while the Cochrane Effective Practice Organization of Care tool will be used for interventional studies. Two reviewers will perform the quality appraisal and a third reviewer will arbitrate any disagreement.

## Data synthesis

If possible, relative risk estimates together with CIs will be calculated from each study, and if the studies are sufficiently homogeneous, a meta-analysis will be performed by pooling estimates across studies using the random-effects approach. Heterogeneity will be examined using the Higgins $I^2$ statistic.[18] Heterogeneity will be considered minimal if $I^2$ values will be between 0% and 30%, moderate if $I^2$ will be between 31% and 50%, and high if $I^2 > 50\%$.

Publication bias will be assessed using Begg's funnel plot and Egger's test. An asymmetrical funnel plot or p value of <0.10 on Egger's test will be considered as an indication of publication bias. If data are sufficient, subgroup analyses will be performed based on the quality of study, age, gender, aetiology of PA, disease severity, diagnostic criteria for PA and ethnicity.

## Patient and public involvement

None.

## DISCUSSION

Once thought a rare disease, PA is nowadays known to be the most frequent form of secondary hypertension, affecting 3–13% of patients with hypertension.[1 19] PA is associated with higher risk of metabolic, renal and cardiovascular complications compared with essential hypertension.[20] The current treatment option for PA includes surgical adrenalectomy or medical therapy with MR antagonists, and the choice of treatment is based on PA subtype classifications and patient preference.[14 21 22] In small clinical studies, surgical treatment has been shown to have better outcomes.[9 10 19] However, large randomised trials are lacking, and no solid conclusion can be drawn.

Previous systematic reviews and meta-analyses performed on this topic[5 14 23 24] have mainly assessed the impact of adrenalectomy versus medical treatment on all-cause mortality and major adverse cardiovascular events (eg, coronary artery disease, stroke, arrhythmia and heart failure) in patients with PA. However, other treatment outcomes, as clinical and biochemical control, quality of life and other comorbidities (rather than cardiovascular) have not been assessed yet.

The comparison between surgical and medical treatment in PA is challenging. Surgical approach has changed over time, with introduction of minimally invasive surgery and robotic approaches that have modified the panorama of the PA treatment, allowing for organ-sparing resection and improved surgical outcomes.[25 26] Therefore, studies published before 2000 will be excluded in the present meta-analysis. Another possible pitfall could be that patients receiving MR antagonists are often undertreated, resulting in an inadequate biochemical and clinical control of the disease.[6 27] In undertreated patients, the plasma renin remains suppressed, which per se results in increased risk of cardiovascular events, renal dysfunction and mortality.[6] On the other hand, adrenalectomy

solves the root cause of PA and, thus, the MR overactivation, making the comparison with medical treatment challenging.[6] It is also important to consider that patients who are treated with surgery may have different disease characteristics compared with those treated medically (eg, unilateral vs bilateral PA or uncertain lateralisation), which may affect treatment outcomes. Additionally, many patients with PA may not be suitable candidates for surgery due to various reasons, such as patient preference or comorbidities. These factors can confound the comparison between surgical and medical therapy for PA; therefore, the interpretation of the results of this comparison should be done with caution, and individualised treatment decisions should be made.[28–30]

Still, no previous study has synthesised the available evidence to date about the effect of different treatment strategies on clinical and/or biochemical control as well as quality of life and renal function. In the present review, we will investigate long-term outcomes in patients with PA who have been treated with either surgery or MR antagonists. In addition, we will gather information on demographic characteristics (age, gender and ethnicity/race) in order to explore their potential impact on treatment outcomes. Overall, we believe that our systemic review will provide valuable information for optimising treatment strategies in patients with PA.

### Ethics and dissemination

As this study is based solely on the published literature, no ethics approval is required. The study findings will be presented at scientific meetings and will be published in an international peer-reviewed scientific journal.

**Author affiliations**
[1]Department of Medicine and Health Science 'V Tiberio', University of Molise, Campobasso, Italy
[2]Department of Internal Medicine and Clinical Nutrition, Institute of Medicine, Sahlgrenska Academy, University of Gothenburg, Gothenburg, Sweden
[3]Department of Endocrinology, Sahlgrenska University Hospital, Gothenburg, Sweden
[4]Biomedical Library, Gothenburg University Library, University of Gothenburg, Gothenburg, Sweden
[5]Department of Surgery, Sahlgrenska University Hospital, Gothenburg, Sweden
[6]Department of Surgery, Institute of Clinical Sciences, Sahlgrenska Academy, University of Gothenburg, Gothenburg, Sweden
[7]Krefting Research Centre, Institute of Medicine, Sahlgrenska Academy, University of Gothenburg, Gothenburg, Sweden
[8]Wallenberg Centre for Molecular and Translational Medicine, Institute of Medicine, University of Gothenburg, Gothenburg, Sweden
[9]Department of Cardiology, Sahlgrenska University Hospital, Gothenburg, Sweden
[10]Department of Molecular and Clinical Medicine, Institute of Medicine at Sahlgrenska Academy, University of Gothenburg, Gothenburg, Sweden

**Acknowledgements** We thank medical librarians Mrs Eva Hessman and Mrs Linda Hammarbäck at the Biomedical Library, Gothenburg University Library, University of Gothenburg, Gothenburg, Sweden for performing the systematic search in the literature and for their valuable feedback on our search strategy and methodology.

**Contributors** OR and DE conceived this study. AR, EG, MCB, EB and DE developed the study protocol and will implement the systematic review under the supervision of AM, BIN and OR. EB and BIN will provide the statistical analysis expertise. EB, BIN and DE will conduct data analysis. EH will develop the search strings and conduct the study search. AR, EG, MCB and DE will take part in the study search, and will perform the screening and extraction of data, whereas AM, BIN, OR and EB will review the work. DE wrote the first manuscript draft, and all authors gave input to the final draft of the protocol.

**Funding** This work received support from the Swedish state under the agreement between the Swedish government and the county councils, the ALF-agreement (ALFGBG-874631).

**Competing interests** None declared.

**Patient and public involvement** Patients and/or the public were not involved in the design, or conduct, or reporting, or dissemination plans of this research.

**Patient consent for publication** Not required.

**Provenance and peer review** Not commissioned; externally peer reviewed.

**ORCID iD**
Daniela Esposito http://orcid.org/0000-0001-8993-2071

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
