## [Reviewer comments · BMJ Open]

ARTICLE DETAILS

TITLE (PROVISIONAL)	Therapeutic Outcomes with Surgical and Medical Management for Primary Aldosteronism: protocol for a Systematic Review and Meta-Analysis
AUTHORS	Rocca, Aldo; Gkaniatsa, Eleftheria; Brunese, Maria Chiara; Hessman, Eva; Muth, Andreas; Nwaru, Bright; Ragnarsson, Oskar; Bobbio, Emanuele; Esposito, Daniela

VERSION 1 – REVIEW

REVIEWER	Gregory Hundemer Ottawa Hospital General Campus, Division of Nephrology
REVIEW RETURNED	16-Feb-2023

GENERAL COMMENTS	Rocca et al present a protocol for a systematic review and meta-analysis on the topic of outcomes with medical vs. surgical management of primary aldosteronism (PA). This is an appropriate topic and I have no concerns about the protocol and study design. I do have a few comments about issues on performing a SR and MA on this topic that I believe may need to be addressed: 1. A problem with comparing outcomes in surgically vs. medically treated PA is that the comparison is confounded by essentially all patients treated surgically having unilateral PA. In contrast, patients treated medically typically have either bilateral PA, unilateral PA where surgery is not pursued for whatever reason (pt not interested, too many comorbidities to pursue surgery, etc.), or PA where lateralization is uncertain. This is a challenge as unilateral vs. bilateral PA can be thought of as 2 different diseases with a shared pathophysiologic pathway. Notably, surgery is not even an option for the majority of PA patients. This can make this comparison of surgery vs. medical therapy a potentially highly confounded one.2. Further, for patients treated with medical therapy for PA, there are a number of different approaches that may be hard to compare. Different providers/studies have different targets with medical therapy (e.g., BP normalization, K normalization, targeting a rise in renin). This may provide significant heterogeneity in the existing literature which may make it a challenge to merge the findings from many of these studies.3. Another issue surrounds the diagnosis of PA, the definition of which varies center-to-center based on ARR and confirmatory testing. Given there is no universal standard definition for PA, how will the authors address this within the SR/MA?
--

REVIEWER	Harini Sarathy University of California San Francisco
REVIEW RETURNED	18-Mar-2023

GENERAL COMMENTS	The proposed systematic review and meta-analysis is well-designed. However, I have a specific concern about the primary aim. 1) The authors specify their primary aim as "The aim of this systematic review is to identify, critically appraise and synthesize available data evaluating whether patients with PA who underwent adrenalectomy have higher survival rate than patients treated with MR antagonists." Their PICO outcomes also compare surgically treated patients with medically treated patients.  - However, in the Discussion section, the authors mention that "it is inaccurate to compare this group of patients (those treated with MRAs) to those treated with adrenalectomy that solves the root cause of PA and may result in better outcomes (27-29)". - Can the authors include "etiology of hyperaldosteronism" (i.e. adrenal adenoma vs adrenal hyperplasia) in the review? A true comparison would be between surgical vs medical treatment of defined adrenal adenomas; since adrenal hyperplasias are not amenable to surgery. Would the authors consider this analysis of surgical vs medical treatment of adrenal adenomas alone as another objective, if they are able to obtain the data? 2) Additional reviews that should be cited:  - A 2021 systematic review and meta-analysis that compared surgical and medical treatments in participants from randomized clinical trials in terms of hypertension control and MACE outcomes should be cited here, and this can help further delineate the objectives of this proposed review. (PMID: 34079522; PMCID: PMC8165438) • A 2022 systematic review and meta-analysis that compares surgical and medical treatment for stroke risk should be cited. (PMID: 36135445; PMCID: PMC9505464) 3) Would add "hyperaldosteronemia" and "aldosteronemia" to the keyword search because of the difference in European 'aemia' and American 'emia' styles.
---

VERSION 1 – AUTHOR RESPONSE

Comments from reviewer #1

Dr. Gregory Hundemer, Ottawa Hospital General Campus

Rocca et al present a protocol for a systematic review and meta-analysis on the topic of outcomes with medical vs. surgical management of primary aldosteronism (PA). This is an appropriate topic and I have no concerns about the protocol and study design.

I do have a few comments about issues on performing a SR and MA on this topic that I believe may need to be addressed:

1. A problem with comparing outcomes in surgically vs. medically treated PA is that the comparison is confounded by essentially all patients treated surgically having unilateral PA. In contrast, patients treated medically typically have either bilateral PA, unilateral PA where surgery is not pursued for whatever reason (pt not interested, too many comorbidities to pursue surgery, etc.), or PA where lateralization is uncertain. This is a challenge as unilateral vs. bilateral PA can be thought of as 2 different diseases with a shared pathophysiologic pathway. Notably, surgery is not even an option for the majority of PA patients. This can make this comparison of surgery vs. medical therapy a potentially highly confounded one.

Answer: Thank you for raising this important issue. We agree with the reviewer that patients who are treated with surgery may have different disease characteristics compared to those treated

medically (e.g. unilateral vs. bilateral PA or uncertain lateralization), which may affect treatment outcomes. Additionally, other factors can confound the comparison between surgical and medical treatment in PA. As pointed out by the reviewer, PA patients may not be suitable candidates for surgery due to various reasons, such as patient preference or comorbidities. Therefore, the interpretation of the results of such comparison should be done with caution, and individualized treatment decisions should be made.

To address this issue, we will however collect data on etiology of PA from the articles included in the systematic review and if we are able to obtain sufficient data, we will perform a sub-analysis of surgical vs medical treatment in different etiology of PA (e.g. unilateral vs. bilateral PA).

We have added these considerations in the manuscript. Thank you for your suggestion. **Please, see the highlighted changes on page 12, lines 246-252 and page 11, line 217.**

2. Further, for patients treated with medical therapy for PA, there are a number of different approaches that may be hard to compare. Different providers/studies have different targets with medical therapy (e.g., BP normalization, K normalization, targeting a rise in renin). This may provide significant heterogeneity in the existing literature which may make it a challenge to merge the findings from many of these studies.

Answer: The reviewer raised a valid point about the potential heterogeneity in the existing literature on medical therapy for PA. It is true that different studies may have different targets for medical therapy, which may make direct comparison challenging. However, systematic reviews and meta-analyses are designed to address these issues by carefully selecting studies that meet predefined criteria and using statistical methods to pool the data from these studies. In addition, as we have described in the protocol, we will assess the heterogeneity among studies using the Higgins I^2 statistic. Thank you for your comment.

3. Another issue surrounds the diagnosis of PA, the definition of which varies center-to-center based on ARR and confirmatory testing. Given there is no universal standard definition for PA, how will the authors address this within the SR/MA?

*Answer: We agree with the reviewer that a high heterogeneity can be found in definition of diagnostic criteria for PA which can have a potential impact on the findings of the meta-analysis. We will address this issue by conducting a thorough and systematic search for diagnostic criteria for PA used in the studies included in the systematic review. These data will be presented in a table showing characteristics of all included studies. If a high heterogeneity is found, a subgroup analysis will be performed based on the different diagnostic criteria used in the included studies to better understand the impact of varying definitions of PA on the overall results. This information has been included in the revised version of the protocol. **Please, see the highlighted changes on page 10, lines 195 and page 11, line 217.***

Comments from reviewer #2

Dr. Harini Sarathy, University of California San Francisco

The proposed systematic review and meta-analysis is well-designed. However, I have a specific concern about the primary aim.

- 1) The authors specify their primary aim as "The aim of this systematic review is to identify, critically appraise and synthesize available data evaluating whether patients with PA who underwent adrenalectomy have higher survival rate than patients treated with MR antagonists." Their PICO outcomes also compare surgically treated patients with medically treated patients.
 - However, in the Discussion section, the authors mention that "it is inaccurate to compare this group of patients (those treated with MRAs) to those treated with adrenalectomy that solves the root cause of PA and may result in better outcomes (27-29)".
 - Can the authors include "etiology of hyperaldosteronism" (i.e. adrenal adenoma vs adrenal hyperplasia) in the review? A true comparison would be between surgical vs medical treatment of defined adrenal adenomas; since adrenal hyperplasias are not amenable to surgery. Would the authors consider this analysis of surgical vs medical treatment of adrenal adenomas alone as another objective, if they are able to obtain the data?

Answer: There are many confounding factors that can affect the comparison between surgical and medical treatment in PA, e.g. etiology of PA, comorbidities, age, patient’s preference etc. The authors think that the comparison is challenging rather than inaccurate and the word “inaccurate” has been changed in the discussion.

We do agree with the reviewer that it is important to gather data on etiology of hyperaldosteronism as it can have a significant impact on treatment strategies and outcomes. We will therefore collect data on etiology of PA from the articles included in the systematic review and if we are able to obtain sufficient data, we will certainly perform a separate analysis of surgical vs medical treatment of adrenal adenomas alone.

Please, see the highlighted changes on page 10, line 195; page 11, line 217; and page 12, line 245.

2) Additional reviews that should be cited:

- A 2021 systematic review and meta-analysis that compared surgical and medical treatments in participants from randomized clinical trials in terms of hypertension control and MACE outcomes should be cited here, and this can help further delineate the objectives of this proposed review. (PMID: 34079522; PMCID: PMC8165438)

• A 2022 systematic review and meta-analysis that compares surgical and medical treatment for stroke risk should be cited. (PMID: 36135445; PMCID: PMC9505464)

*Answer: The systematic reviews and meta-analyses suggested by the review have been included in the revised version of the protocol. These reviews have mainly assessed all-cause mortality and the rate of major adverse cardiovascular events (e.g. coronary artery disease, stroke, arrhythmia, and heart failure) after adrenalectomy vs medical treatment in PA patients. However, the effect of these treatments on clinical and biochemical control and other comorbidities (rather than cardiovascular morbidity) and quality of life has not been investigated yet. Our systematic-review will be the first review to assess these outcomes. These considerations have been included in the protocol. Thank you for this suggestion. **Please, see the highlighted changes on page 11, lines 230-235.***

3) Would add “hyperaldosteronemia” and “aldosteronemia” to the keyword search because of the difference in European ‘aemia’ and American ‘emia’ styles.

Answer: As suggested by the reviewer, we have performed a new search with the suggested terms added in the search strategy (the search was performed on the 6th of April 2023). One additional article was retrieved, but it was excluded because it was published in 1986 and date of publication before 2000 is an exclusion criterion. Therefore, adding alternative spelling for the terms hyperaldosteronemia[tiab] AND aldosteronemia[tiab] (i.e. hyperaldosteronaemia[tiab] OR aldosteronaemia[tiab]) to the search strategies, does not retrieve additional search results for the actual time frame. Hence, the terms are omitted. Thank you for this suggestion.

VERSION 2 – REVIEW

REVIEWER	Gregory Hundemer Ottawa Hospital General Campus, Division of Nephrology
REVIEW RETURNED	11-Apr-2023
GENERAL COMMENTS	The authors have adequately responded to all prior comments.